# Laser Stripe Centerline Extraction Method for Deep-Hole Inner Surfaces Based on Line-Structured Light Vision Sensing

**DOI:** 10.3390/s25041113

**Published:** 2025-02-12

**Authors:** Huifu Du, Daguo Yu, Xiaowei Zhao, Ziyang Zhou

**Affiliations:** 1School of Mechanical Engineering, North University of China, Taiyuan 030051, China; b20230211@st.nuc.edu.cn (H.D.); b20220206@st.nuc.edu.cn (X.Z.); 2202041109@st.nuc.edu.cn (Z.Z.); 2School of Mechanical Engineering, Nantong Institute of Technology, Nantong 226001, China

**Keywords:** laser stripe centerline extraction, image processing, minimum spanning tree, depth-first search, noise removal

## Abstract

This paper proposes a point cloud post-processing method based on the minimum spanning tree (MST) and depth-first search (DFS) to extract laser stripe centerlines from the complex inner surfaces of deep holes. Addressing the limitations of traditional image processing methods, which are affected by burrs and low-frequency random noise, this method utilizes 360° structured light to illuminate the inner wall of the deep hole. A sensor captures laser stripe images, and the Steger algorithm is employed to extract sub-pixel point clouds. Subsequently, an MST is used to construct the point cloud connectivity structure, while DFS is applied for path search and noise removal to enhance extraction accuracy. Experimental results demonstrate that this method significantly improves extraction accuracy, with a dice similarity coefficient (DSC) approaching 1 and a maximum Hausdorff distance (HD) of 3.3821 pixels, outperforming previous methods. This study provides an efficient and reliable solution for the precise extraction of complex laser stripes and lays a solid data foundation for subsequent feature parameter calculations and 3D reconstruction.

## 1. Introduction

Deep holes are defined as holes with a depth-to-diameter ratio of greater than 5. Deep hole inspection technology must overcome challenges such as hole depth, insufficient illumination, complex geometries, and high-precision measurement. Therefore, the difficulty of deep hole inspection is significantly higher than that of external surface inspection [1]. Deep hole inspection based on the annular light-sectioning method is one of the most important applications of line-structured light sensors. This technology is widely used in the aerospace, automotive manufacturing, precision machining, petroleum drill pipe, and defense industries. It offers advantages such as high precision, fast measurement, and non-contact inspection [2,3]. The core technology involves projecting 360° line-structured light onto the inner wall of the deep hole to form continuous laser stripes. High-precision vision sensors capture these stripe images, and efficient image processing algorithms are then used to accurately extract the centerline coordinates of the stripes. Based on these coordinate data, further measurements of geometric parameters such as inner diameter, roundness, straightness, and cylindricity can be performed, as well as 3D reconstruction. Throughout the measurement process, extracting the laser stripe centerline is critical, as its accuracy directly affects the reliability and precision of the measurement results [4,5].

The measurement accuracy of deep hole components is crucial for ensuring the assembly precision and functional reliability of the parts, especially in the measurement of complex deep hole components such as internal gears, internal splines, and gun barrels. The complexity of geometric shapes, surface roughness, machining tool marks, and surface reflection characteristics often cause discontinuous and uneven laser stripes, significantly increasing the difficulty of extracting the laser stripe centerline [6]. These factors pose severe challenges to traditional image processing methods, such as the Steger, Canny, Sobel, Prewitt, and Roberts algorithms [7,8,9]. When dealing with complex geometric stripe images, contour point cloud data extracted by these algorithms often contain burrs and low-frequency random noise, making it difficult for them to meet the high-precision measurement requirements in terms of extraction accuracy and reliability [10].

In the study of laser stripe centerline extraction, many researchers have improved and optimized existing algorithms to enhance their adaptability and accuracy. Common improvement methods include Hessian matrix-based algorithms, grayscale gravity methods, DBSCAN clustering, adaptive bidirectional grayscale methods, and adaptive convolution techniques. These methods each have their own advantages and limitations in different application scenarios. Hessian matrix-based algorithms [11,12] identify the stripe center position by calculating the second-order derivatives of the image. They are suitable for flat or relatively simple surfaces but are highly sensitive to noise and perform poorly when dealing with complex stripes or blurry edges. The grayscale gravity method [13] optimizes stripe extraction using image grayscale information and a gravity model, effectively handling flat stripes, but its accuracy decreases on complex surfaces and in noisy environments. DBSCAN clustering [14], as an unsupervised algorithm, effectively handles noise and non-uniform stripe data, offering strong adaptability. However, its computational efficiency is low in dense stripe regions or large-scale datasets, limiting its accuracy. The adaptive bidirectional grayscale method [15] incorporates stripe contextual information, making it suitable for irregular stripe shapes, but it has high computational complexity and poor real-time performance when processing large-scale stripes. Adaptive convolution techniques [16,17,18], particularly convolutional neural network (CNN)-based methods, dynamically adjust extraction strategies to improve accuracy. They perform well for complex backgrounds and discontinuous stripe processing but require a large amount of labeled data and consume substantial computational resources.

In recent years, emerging technologies have made significant progress in improving the accuracy and adaptability of laser stripe extraction. For example, deep learning-based image segmentation methods such as U-Net [19] can effectively handle complex backgrounds and irregular stripe patterns, enhancing extraction accuracy by learning image structural information. Adaptive filtering methods [20] dynamically adjust filter parameters, performing well in noisy environments and on complex surfaces. Multi-scale analysis methods [21] combine stripe features at different resolutions, showing strong adaptability, especially when processing multi-scale stripes. Deep generative adversarial networks (GANs) [22] have been widely applied to generate high-quality stripe images, demonstrating remarkable advantages in noise removal and stripe restoration. Self-supervised learning [23] enables effective training without requiring large amounts of labeled data, improving algorithm scalability and flexibility. Sensor fusion techniques [24,25], which integrate multiple sensor sources such as laser stripes, visual imaging, and structured light, effectively address challenges posed by complex surfaces and diverse materials, providing more precise 3D reconstruction and stripe extraction results. In particular, the combination of deep learning and sensor fusion [26,27] offers significant advantages over traditional methods in noise suppression, background complexity handling, and real-time performance. However, despite these advancements, these methods still face challenges in applications involving complex components due to their high computational demands and relatively low real-time performance, necessitating further research and optimization.

The research team previously used the annular light-sectioning method to measure the diameter of deep-hole steel pipe components. For components with relatively smooth inner surfaces, the Steger algorithm can quickly and accurately extract the contour of the annular laser beam. However, when applying this algorithm to extract the laser stripe centerline of components with complex internal cavity shapes (e.g., petal-shaped structures, internal gears, splines, and internal octagons), issues such as burrs, low-frequency noise, and redundant branches often arise. These problems severely affect the accuracy and completeness of the extraction results, making it difficult to meet the high-precision measurement and complex component analysis requirements [28]. Nevertheless, the Steger algorithm can still retain contour details during the overall extraction process. However, automatically identifying the precise centerline of laser stripes from noisy and sub-pixel point cloud data remains a major challenge in the field of complex laser stripe extraction [29]. Currently, noise points are mainly removed manually to obtain a complete contour before performing feature parameter calculations and 3D reconstruction. This process is time-consuming and susceptible to human error.

Against this background, this paper proposes a method for extracting the laser stripe centerline in complex deep-hole internal cavities based on the minimum spanning tree (MST) and depth-first search (DFS). This study first employed the Steger algorithm to generate a sub-pixel point cloud model that contains noise. Then, MST and DFS are applied to search and track the main contour, effectively eliminating interfering point cloud data to accurately obtain the point cloud coordinates of the laser stripe’s geometric center. This method fully utilizes MST’s advantage in constructing an effective connectivity structure for point cloud data, ensuring reasonable connections between point clouds. By combining DFS’s traversal and path-searching capabilities, it precisely locates and tracks the main contour. Experimental results show that this method can quickly and effectively remove burrs and extraneous branches, significantly improving the extraction accuracy of laser stripe centerlines in complex components. This provides a reliable data foundation for subsequent feature parameter calculations, 3D reconstruction, and further analysis [30].

### Technical Terms and Abbreviations

To facilitate understanding, the following abbreviations and technical terms are used in this paper:

MST (minimum spanning tree): An algorithm used to construct a connected, acyclic graph that connects all vertices while minimizing the total edge weight.

DFS (depth-first search): An algorithm used to traverse or search tree or graph structures, starting from one node and exploring as deep as possible before backtracking.

DSC (dice similarity coefficient): A statistical measure used to quantify the similarity between two sample sets, with values ranging from 0 to 1, where 1 represents perfect overlap and 0 indicates no intersection.

HD (Hausdorff distance): A measure used to assess the maximum distance between two sets, often used to evaluate the maximum deviation between two boundaries.

## 2. Analysis of Laser Stripe Characteristics in Complex Deep-Hole Geometries

In the field of image processing, HALCON is a commercially available computer vision software library (https://www.mvtec.com/products/halcon, accessed on 10 February 2025) that has been widely validated through industrial applications, demonstrating excellent reliability and efficiency. The linesgauss operator in HALCON, based on the core of the Steger algorithm, is capable of effectively detecting various geometric shapes, such as straight lines, curves, arcs, elliptical arcs, and waveforms. This algorithm is an efficient line feature extraction method, particularly outstanding in applications that require high precision and subpixel-level edge detection. In the previous research, the Steger algorithm was successfully applied to the point cloud extraction of the laser stripe centerline on the inner surface of a deep hole part, and significant results were achieved. Therefore, this study first applies the Steger algorithm to extract the laser stripe centerlines from contours of varying complexities, aiming to investigate the performance of the algorithm in such applications.

The research team previously conducted a systematic analysis of the deep hole inspection principles based on the annular light-sectioning method. They successfully developed deep hole inspection equipment and completed the 3D reconstruction of steel pipe components, as well as temperature error compensation [31,32]. The laser stripe images presented below were all acquired using the deep hole inspection equipment independently developed by the research team.

Figure 1a shows the laser stripe image of a smooth deep hole, while Figure 1b presents a 3D intensity distribution of the grayscale values from the image. It is evident from the figure that the laser stripe is uniformly distributed and has a simple contour shape. Figure 1c shows the result of the laser stripe centerline extraction using the Steger algorithm, and Figure 1d is a local zoomed-in view of the extracted result. From these results, it is clear that the Steger algorithm performs well in extracting the centerline of laser stripes with simple contours. The algorithm accurately extracts a subpixel contour curve, facilitating subsequent contour feature parameter calculations and 3D reconstruction.

Figure 2a shows an image of the contour of the flap laser stripe, which exhibits an annular distribution consisting of a combination of straight lines and circular arcs. Figure 2b, on the other hand, shows the three-dimensional distribution of the gray intensity of this image. It can be observed that there is an obvious abrupt change in the gray intensity at the junction of straight lines and circular arcs, which leads to an uneven energy distribution of the laser stripes. Figure 2c demonstrates the results of the laser stripe centerline extracted using Steger’s algorithm (after optimizing the parameters), and Figure 2d is a local zoomed-in view of the laser stripe center extraction results. It can be clearly seen that the algorithm identifies multiple line segments, and the figure distinguishes the extracted line segments with different colors, which is again very different from the extraction result of the laser stripe centerline of the smooth hole in Figure 2c above, and there are many burr noises in the junction of the straight line and the circular arc. This kind of gray scale mutation easily causes Steger’s algorithm to misidentify the pseudo-edges, which poses a challenge to the extraction of the centerline.

The inner surface of an internal gear has a complex shape and can be considered one of the representatives for the extraction of complex laser stripe centerlines. Figure 3a shows the grayscale image of the laser stripes on the inner surface of the internal gear. It can be observed that the laser stripe intensity distribution is uneven, with alternating light and dark areas, which significantly differs from the laser stripe grayscale image of the smooth deep hole surface. Figure 3b presents the 3D intensity distribution of the laser stripes on the internal gear. From the figure, it is clear that there is a significant grayscale intensity difference between the tooth crest, tooth root, and the involute surface. The laser stripe intensity is higher on the tooth crest and tooth root, while the intensity on the involute surface is relatively lower. The main reason for this phenomenon lies in the fact that the optical flux generated by the annular laser beam is uniform within a unit angle. However, within the same angular range, the length of the involute curve is greater than that of the tooth crest and tooth root curves, resulting in lower optical intensity per unit length on the involute curve, and higher intensity on the tooth crest and tooth root curves. Additionally, the variation in laser stripe grayscale intensity is influenced by factors such as surface roughness, machining tool marks, and surface reflectivity.

Figure 3c shows the result of the laser stripe centerline extraction on the internal gear using the Steger algorithm, and Figure 3d provides a local zoom-in of the extracted result. It can be clearly seen that the algorithm identifies multiple line segments, distinguishing them by different colors. However, in the zoomed-in subpixel point cloud, numerous noise points can be observed, appearing as burrs or chaotic branch points. Despite this, the overall extraction effectively retains the detailed information of the internal gear contour. By manually removing the noise points, a complete contour of the internal gear can be obtained.

Figure 4a shows the grayscale image of the laser stripes on the surface of a rectangular spline. From the image, it can be observed that the laser stripe intensity distribution is similar to that of the internal gear surface. Figure 4b displays the 3D intensity distribution of the laser stripes on the rectangular spline. It is clearly evident that the laser stripe intensity is higher in the arc regions, while the intensity is relatively lower in the rectangular keyway areas. This phenomenon is similar to the behavior observed in the internal gear laser stripes and is mainly influenced by a combination of factors, including the inner surface profile shape, surface roughness, machining tool marks, and surface reflectivity. Figure 4c presents the result of laser stripe centerline extraction on the rectangular spline using the Steger algorithm, and Figure 4d shows a local zoom-in of the extracted result. From the images, it is clear that the algorithm successfully identifies multiple line segments, with different colors representing different segments. However, in the zoomed-in subpixel point cloud, a large number of noise points can be observed, which primarily appear as burrs or chaotic branch points. Despite this, the overall extraction process retains the detailed information of the rectangular spline contour quite effectively.

Figure 5 presents the feature analysis and geometric centerline extraction results of the internal octagonal laser stripe image. Figure 5a shows the grayscale image of the internal octagonal laser stripes, where the stripe distribution exhibits a regular octagonal pattern, with the stripe intensity more evenly distributed along the edges. Figure 5b presents the 3D grayscale distribution of the image. From this, it is clear that the grayscale intensity along one edge of the octagon is higher, while the intensity along the other edges remains relatively uniform, with a few points showing sudden intensity changes. This distribution characteristic is closely related to surface shape, roughness, machining tool marks, and reflective properties. Figure 5c displays the result of the geometric centerline extraction of the internal octagonal laser stripes based on the Steger algorithm, with different colors used to distinguish each line segment. The overall extraction result is relatively clear. Figure 5d provides a local zoom-in of the extracted result, where the details of the local line segments can be clearly observed. However, at the junctions of these segments, multiple line segments are identified, with numerous noise points and chaotic branches appearing.

In summary, for images with simple shape contours and uniform energy intensity distribution, the mature Steger algorithm can accurately extract the centerline of the laser stripes. However, when dealing with complex laser stripe images, the limitations of this algorithm become apparent. These parts, due to their complex inner surface shapes, varying roughness, random machining tool marks, diverse reflective properties, and the combined effects of laser and sensor system errors, result in low-frequency random noise signals in the subpixel contour point cloud data extracted by the Steger algorithm. This error is difficult to fully eliminate. The main issues with the Steger algorithm in complex laser stripe centerline extraction are as follows.

Shape Complexity: Complex contours, such as petals, internal gears, rectangular splines, etc., include multiple concave and convex structures, and the internal structural differences may lead to uneven illumination, thereby reducing image contrast and significantly affecting the extraction results.

Noise Interference: Reflections from the metal surface or machining marks are easily misidentified as edges, increasing the difficulty and complexity of stripe recognition.

Curvature Variation: At the junctions of straight lines with straight lines, straight lines with arcs, or arcs with arcs, large curvature variations are common, which can cause discontinuities in the extraction results or generate spurious edges.

Parameter Adjustment Limitations: Although the Steger operator can improve extraction results through parameter adjustments, for complex shapes such as petals, internal gears, rectangular splines, and internal octagons, relying solely on parameter adjustments cannot achieve ideal detection outcomes.

These limitations significantly affect the direct application of the Steger algorithm in complex laser stripe centerline extraction. However, from an overall extraction perspective, the Steger algorithm still retains the contour’s detailed information relatively well. Currently, complete contours are obtained by manually removing noisy points from point cloud data containing noise, which is a time-consuming and human-influenced process. Therefore, this study proposes an efficient method to automatically extract a complete contour curve from point cloud data containing noise points, providing an effective solution for the precise extraction of complex contours.

## 3. Methodology

Based on the above laser stripe extraction results, it can be seen that the Steger algorithm has some limitations in dealing with complex laser stripe centerlines. Nevertheless, it can be observed from the extraction results of the algorithm that the details of the laser stripes are still well preserved. However, the extracted sub-pixel point cloud contains a large number of noise points and cluttered branching lines. Therefore, we can post-process the sub-pixel point cloud of laser stripes extracted by Steger’s algorithm, and use mathematical methods to quickly eliminate the noise points in the point cloud, so as to obtain accurate geometric center point cloud data. To this end, this study proposes a method combining minimum spanning tree and depth-first search for effective data processing of point clouds containing noise points.

### 3.1. Minimum Spanning Tree (MST)

A minimum spanning tree is a subgraph of an undirected weighted graph that connects all vertices and has no rings, and whose total edge weights are minimized. In this study, the coordinates of a sub-pixel point cloud containing noise points can be viewed as an undirected weighted subgraph and the minimum spanning tree can be realized by the greedy Prim algorithm. The algorithm starts from a starting node and expands gradually, connecting the unvisited nodes by selecting edges with the minimum weight, where an edge is a connecting line between two points.

Mathematically, let G=(V,E) be a connected undirected graph, where V is the set of vertices, E is the set of edges, and W is the weight function for the edges. The goal is to find the minimum spanning tree T of G, where the edge set of E(T)⊆E, satisfies the following:➢T is a connected and acyclic subgraph.➢|E(T)|=|V|−1.➢The total weight $W(T)=\sum_{e\in E(T)}w(e)$) is minimized.


The steps for generating the MST are as follows.

Input variables:

Select an initial vertex $v_{0}\in V$.Define the set $A=\{v_{0}\}$ to represent the visited vertices.Define the set E(T)=∅ represent the edges of the spanning tree.

Algorithm body:

While A≠V, repeat the following steps:Choose an edge (u,v)∈E in E such that u∈A and v∈V∖A, and w(u,v) is minimal.Add edge (u,v) to E(T), i.e., E(T)=E(T)∪{(u,v)}.Add vertex v to the set A, i.e., A=A∪{v}.

Output variables:

When A=V, (V,E(T)) is the minimum spanning tree.The total weight is $W(T)=\sum_{e\in E(T)}w(e)$

As shown in Figure 6, four sets of random points are generated in a 2D plane, each consisting of 9 points labeled: A, B, C, D, E, F, G, H, and I. The Euclidean distance is used to calculate the distance between each pair of points, and the numbers on one side of each line represent the edge weights. This forms a weighted connected graph. The Prim algorithm [33] is then used to construct the MST from the weighted connected graph. The red line segments in the figure represent the MST, where all vertices are connected, and the total edge length between all points is minimized. The process involves gradually selecting the edge with the smallest weight at each step, ensuring that each connected vertex is part of the MST.

### 3.2. Depth-First Search (DFS)

Depth-first search (DFS) is an algorithm for traversing a search tree or graph [34]. The algorithm starts from a starting node and visits nodes as deeply as possible along a path until it reaches a node where it is not possible to proceed further, and then backtracks to the previous node to continue exploring other paths. This process is carried out recursively by visiting deeper and deeper sub-nodes until the entire sub-graph has been traversed. Since MSTs are acyclic connected trees where the path from one node to another is unique, DFS can be used to find the path from the starting node to the terminating node in a minimum spanning tree. As shown in Figure 7, the maximum and minimum value points in the *Y*-axis direction are selected as the start and termination points in the MST, and the unique path from the start point to the termination point found using DFS is shown in Figure 7, where the green line segment indicates the path searched by the DFS’s in the MTS, which demonstrates its unique advantage in exploring complex structures.

In this study, we employ a MST to construct the basic connectivity structure of the point cloud data. The acyclic nature of the MST ensures the existence of unique paths between points. Based on this, we apply a DFS algorithm to traverse the entire tree structure from a specified start node until reaching a predetermined termination node to identify and eliminate points that do not match the expected path. Through four sets of random cases, we thoroughly explore a path search method that combines the principles of minimum spanning tree (MST) and depth-first search (DFS). Simulation results indicate that this method can effectively remove spurious and cluttered branch interference points from point cloud data containing noise points and successfully identify the unique path between the start and end points. Therefore, this method is suitable for extracting the centerline of complex laser stripes.

## 4. Experiment and Analysis

In Section 3 of this paper, we detailed a complex laser stripe centerline extraction method based on the minimum spanning tree (MST) and depth-first search (DFS). This method was initially verified for feasibility using four sets of random data. In this section, we apply the proposed algorithm to extract laser stripe centerlines for complex shapes, including petal-shaped, internal gear, rectangular spline, and internal octagonal patterns. To better present the results, this study uses a segmented approach to display the extraction outcomes. The specific steps of the overall algorithm are as follows:➢Convert the sub-pixel point cloud coordinates to relative positions with respect to a specific origin and calculate the polar coordinates (angle) for each point.➢Divide the angle range into multiple segments (for example, each 45° as one segment) for segmented processing.➢For each segment, extract the points within the specified angle range and compute the Euclidean distance between each pair of points to construct the distance matrix.➢Use the Prim algorithm to construct the MST from the distance matrix, ensuring that within each angular segment, all points are connected with the minimum total edge length.➢For each angular segment, select the starting and ending points: the point closest to the starting angle of the segment is chosen as the starting point. This is performed by calculating the absolute difference between each point’s angle and the starting angle and selecting the point with the smallest difference as the starting point. Similarly, the point closest to the ending angle is selected as the ending point.➢Find the path from the start point to the end point on a minimal spanning tree using DFS and visualize it.

By following these steps, the extracted complex laser stripe centerlines for different shapes are obtained, as shown in Figure 8, Figure 9, Figure 10 and Figure 11. To better demonstrate the advantages of the MST and DFS-based method in extracting complex laser stripe centerlines, we compare the extraction results with the original point cloud coordinates. The blue area in the figure shows the subpixel point cloud data extracted using the Steger algorithm, which contains a significant amount of burrs and scattered branch noise points. The red area, on the other hand, represents the laser stripe center coordinates automatically extracted using the MTS and DFS algorithms. This indicates that the proposed algorithm can effectively remove noise and form an accurate and complete contour. After performing multiple experiments on 32 different shape samples, the results indicate that the proposed MST and DFS-based method excels in extracting centerlines from complex laser stripes. This method can effectively remove spurious and erratic branch points from point clouds containing noise, ensuring that the extracted centerlines are coherent and smooth. These results validate the stability and practicality of the algorithm.

Dice Similarity Coefficient (DSC):

The dice similarity coefficient (DSC) is a statistical measure used to assess the similarity between two sample sets [35]. Its value ranges from 0 to 1, where 1 indicates perfect overlap and 0 indicates no intersection. In this study, the DSC is used to compare the overlap between the centerline extraction results based on MST and DFS with the manually extracted results. The formula is as follows:(1)$$DSC=\frac{2\times |X\cap Y|}{\left|X|+|Y\right|}$$

In Formula (1), X and Y represent the pixel sets of the automated segmentation and the manual segmentation, respectively.

Hausdorff Distance (HD):

The Hausdorff distance (HD) is a metric used to measure the maximum distance between two sets. It quantifies the greatest deviation between the boundaries of the two sets. In this study, the HD is used to evaluate the maximum distance between the boundary of the laser stripe centerline extracted using the MST and DFS methods and the boundary of the manually extracted centerline. It is defined as(2)$$HD\left(X,Y\right)=max\left\{\underset{x\in X}{sup}\;\underset{y\in Y}{inf}\;d\left(x,y\right),\underset{y\in Y}{sup}\;\underset{x\in X}{inf}\;d\left(x,y\right)\right\}$$

In Formula (2), d(x, y) is the Euclidean distance between points x and y, and sup and inf represent the supremum and infimum, respectively.

This study evaluated the performance of laser stripe centerline extraction for four different test objects: petal shape, internal gear, spline, and internal octagon. Figure 12 shows the laser stripe centerline extraction results based on MTS and DFS methods, alongside the manually extracted results. The first row presents the automatic extraction results using MTS and DFS, while the second row shows the manually extracted results. As can be seen from the figure, the extraction results based on the MTS and DFS methods are highly similar to the manually extracted results. This similarity is further demonstrated in the evaluation using the DSC and HD, with the specific values shown in Table 1.

The evaluation metrics indicate that the DSC values for all test objects are very close to 1, with specific values of 0.9986 for petal shape, 0.9987 for internal gear, 0.9953 for spline, and 0.9992 for internal octagon. These results demonstrate that the overlap between the laser stripe centerline extraction results based on MTS and DFS methods and the manually extracted results is extremely high, almost identical. This high consistency reflects the universality and accuracy of the algorithm across different shapes. The maximum HD value is 3.3821 pixels for the petal shape, with other objects showing HD values of 1.6414 for internal gear, 2.0000 for spline, and 0.9653 for internal octagon. These results suggest that the maximum boundary deviation between the MTS and DFS extraction results and the manually extracted results is minimal, confirming the effectiveness of the algorithm in precise localization and boundary preservation.

Combining the results of DSC and HD, it can be concluded that the MTS and DFS-based laser stripe centerline method demonstrates extremely high accuracy and consistency across test objects with various shapes and sizes. The DSC values close to 1 indicate that the extraction results are almost identical to the manual reference, while the low HD values further validate the algorithm’s excellent performance in boundary accuracy. Together, these metrics demonstrate the good adaptability and robustness of the proposed method for center extraction of complex laser fringes.

## 5. Discussion and Conclusions

In this study, we approach the problem of laser stripe centerline extraction from the perspective of path planning and propose a mathematical method based on minimum spanning tree (MTS) and depth-first search (DFS). The advantage of this method lies in its ability to search and track the main contour along the target area in a point cloud model containing noise, effectively eliminating spurious, disordered branches of interference data, and accurately identifying the geometric coordinates of the laser stripe center.

To validate this method, we employed a deep hole detection device developed in our previous work to collect laser stripe images from the inner walls of several different complex deep hole parts (such as petal-shaped, internal gears, rectangular splines, and internal octagons), and performed centerline extraction of the complex laser stripes. The similarity and differences between the laser stripe extraction results based on MTS and DFS and those from manual extraction were evaluated using dice similarity coefficient (DSC) and Hausdorff distance (HD). The experimental results show that the point cloud post-processing method based on MTS and DFS achieves a DSC value close to 1 and a maximum Hausdorff distance of 3.3821 pixels, which is highly similar to the manually selected point cloud data. Compared with other methods, the crack propagation method [36] shows an average DSC and HD of 0.9387 and 1.6290 mm, respectively; “Real-Time Ultrasound Segmentation” [37] has an average DSC and HD of 0.9191 and 6.4700 mm; and “Knowledge-based Grouping Adaptation and New Step-wise Registration with Discrete Cosines” [38] has an average DSC of 0.8500. These results indicate that the laser stripe extraction method based on MTS and DFS outperforms others in terms of DSC and HD, exhibiting higher precision and better performance, thus fully demonstrating its robustness and adaptability in complex laser stripe centerline extraction.

This paper addresses the challenges of extracting the laser stripe centerline in complex deep hole inner cavities and proposes an innovative method based on the minimum spanning tree (MST) and depth first search (DFS). By combining the point cloud connectivity construction capability of MST and the path search advantage of DFS, the method effectively eliminates noise points and scattered branches, achieving the high-precision extraction of laser stripe centerlines in complex shapes. This study not only provides new technological means for high-precision measurement of complex deep hole inner cavities but also offers valuable reference for image processing and 3D reconstruction tasks in related fields. Future work should further explore the potential application of this method using larger-scale data and in more complex environments, as well as its integration with other advanced technologies to further enhance its performance and applicability.

## Figures and Tables

**Figure 1 sensors-25-01113-f001:**
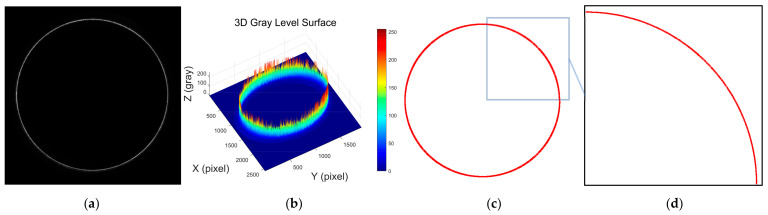
Analysis of laser stripe image features and geometric centerline extraction in a smooth deep hole. (**a**) Contour image of the laser stripe in a smooth deep hole; (**b**) 3D distribution of grayscale intensity in the laser stripe image of a smooth deep hole; (**c**) extracted laser stripe centerline in a smooth deep hole using the Steger algorithm; (**d**) locally magnified view of the extracted laser stripe centerline in a smooth deep hole.

**Figure 2 sensors-25-01113-f002:**
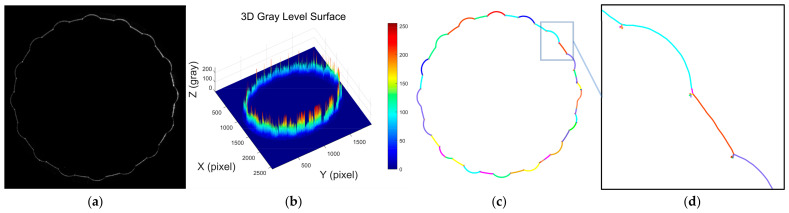
Analysis of laser stripe image features and geometric centerline extraction in petal-shaped structures. (**a**) Contour image of the laser stripe in a petal-shaped structure; (**b**) 3D distribution of grayscale intensity in the laser stripe image of a petal-shaped structure; (**c**) extracted laser stripe centerline in a petal-shaped structure using the Steger algorithm; (**d**) locally magnified view of the extracted laser stripe centerline in a petal-shaped structure.

**Figure 3 sensors-25-01113-f003:**
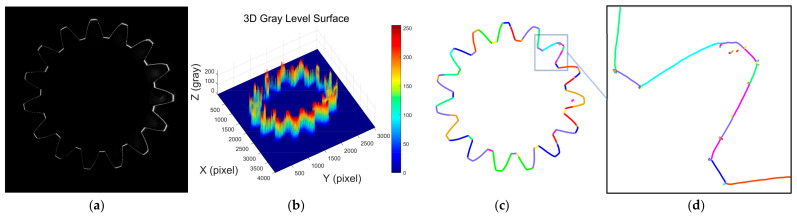
Analysis of laser stripe image features and geometric centerline extraction in internal gears. (**a**) Contour image of the laser stripe in an internal gear; (**b**) 3D distribution of grayscale intensity in the laser stripe image of an internal gear; (**c**) extracted laser stripe centerline in an internal gear using the Steger algorithm; (**d**) locally magnified view of the extracted laser stripe centerline in an internal gear.

**Figure 4 sensors-25-01113-f004:**
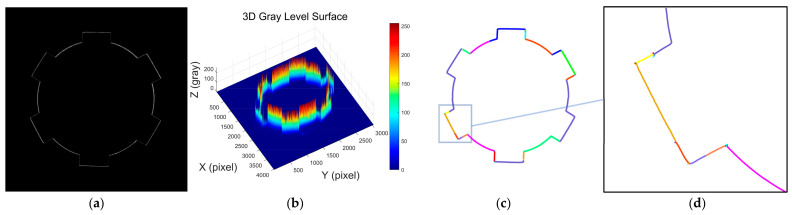
Analysis of laser stripe image features and geometric centerline extraction in rectangular splines. (**a**) Contour image of the laser stripe in a rectangular spline; (**b**) 3D distribution of grayscale intensity in the laser stripe image of a rectangular spline; (**c**) extracted laser stripe centerline in a rectangular spline using the Steger algorithm; (**d**) locally magnified view of the extracted laser stripe centerline in a rectangular spline.

**Figure 5 sensors-25-01113-f005:**
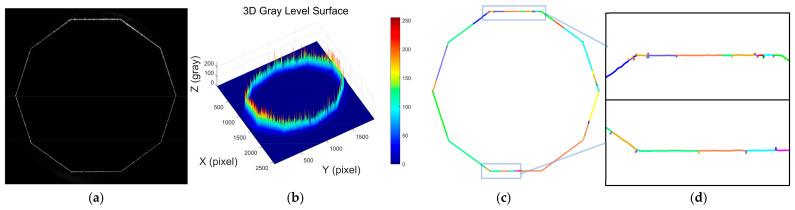
Analysis of laser stripe image features and geometric centerline extraction in internal octagons. (**a**) Contour image of the laser stripe in an internal octagon; (**b**) 3D distribution of grayscale intensity in the laser stripe image of an internal octagon; (**c**) extracted laser stripe centerline in an internal octagon using the Steger algorithm; (**d**) locally magnified view of the extracted laser stripe centerline in an internal octagon.

**Figure 6 sensors-25-01113-f006:**
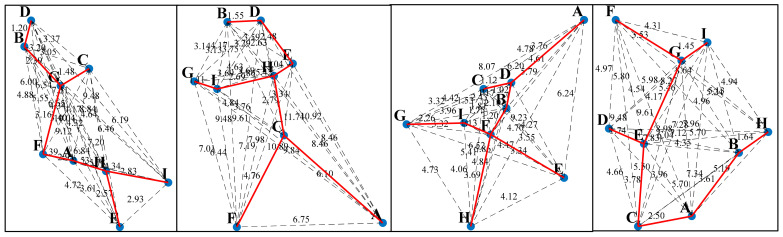
Minimum spanning tree generation example.

**Figure 7 sensors-25-01113-f007:**
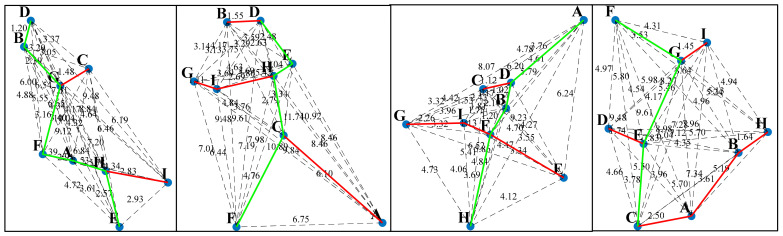
DFS path search example in MST.

**Figure 8 sensors-25-01113-f008:**
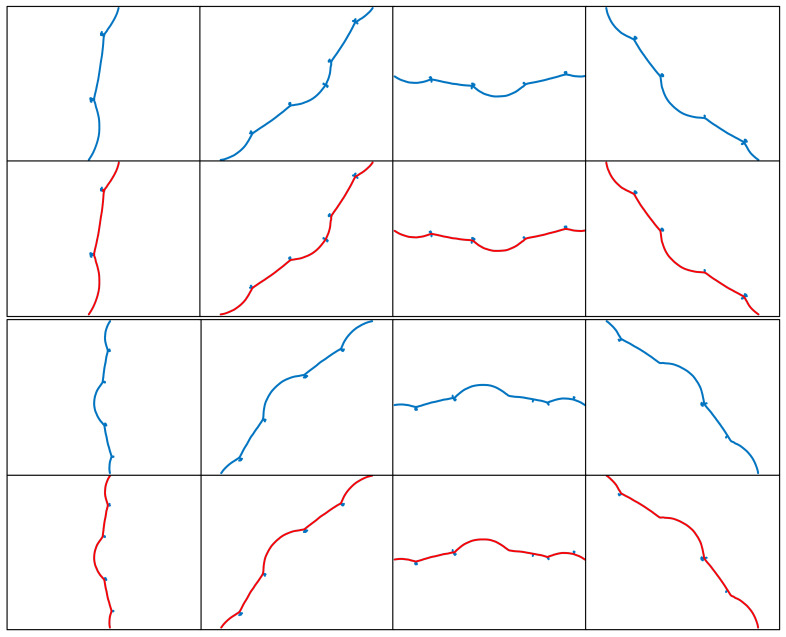
Extraction results of the petal-shaped laser stripe centerline.

**Figure 9 sensors-25-01113-f009:**
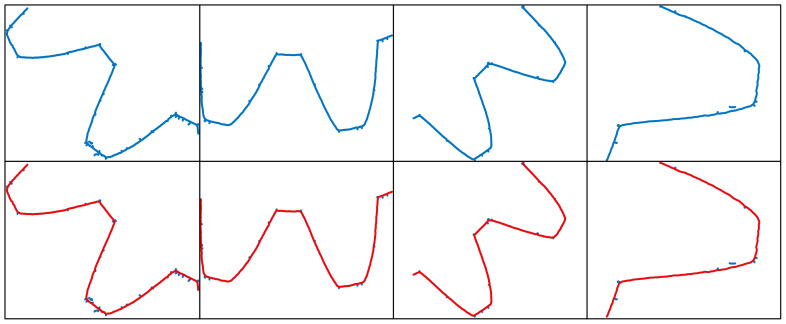
Extraction results of the internal gear laser stripe centerline.

**Figure 10 sensors-25-01113-f010:**
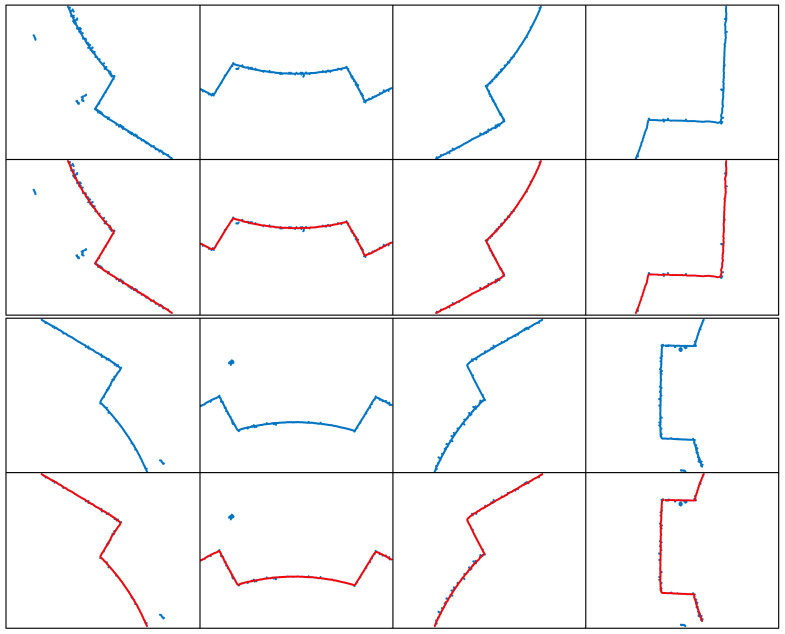
Extraction results of the rectangular spline laser stripe centerline.

**Figure 11 sensors-25-01113-f011:**
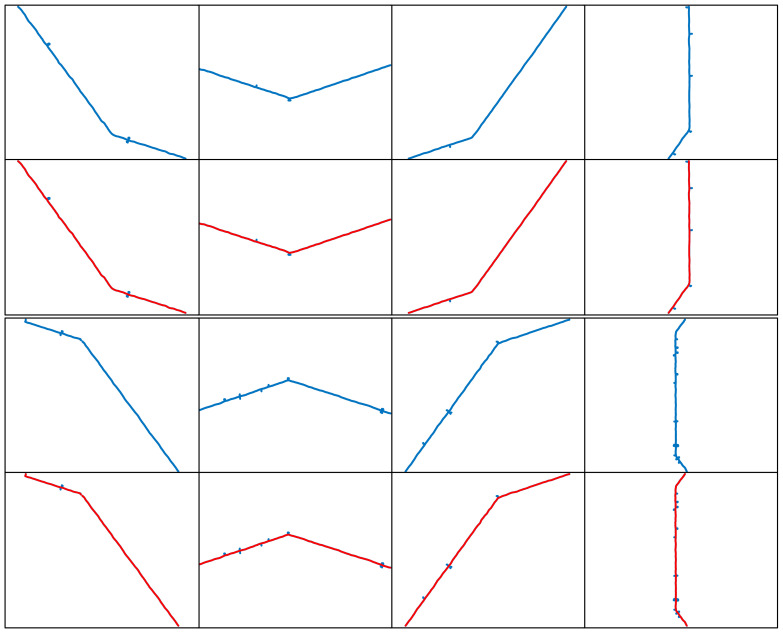
Extraction results of the internal octagonal laser stripe centerline.

**Figure 12 sensors-25-01113-f012:**
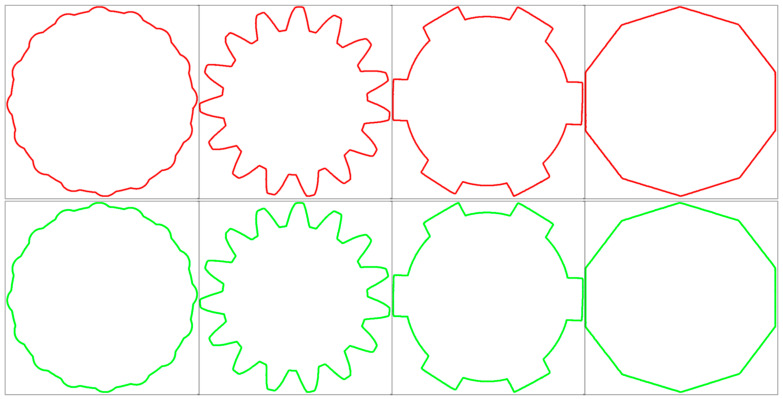
Laser stripe centerline extraction results based on MST and DFS compared with manual extraction results.

**Table 1 sensors-25-01113-t001:** DSC and HD numerical table of different types of laser stripes.

Laser Stripe	Petal-Shaped	Inner Gear	Spline	Internal Octagon
Hausdorff Distance	3.3821	1.6414	2.0000	0.9653
Dice Similarity Coefficient (pixel)	0.9986	0.9987	0.9953	0.9992

## Data Availability

Data are contained within the article.

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
