# Peer review of "Laser Stripe Centerline Extraction Method for Deep-Hole Inner Surfaces Based on Line-Structured Light Vision Sensing"

_sensors, 2025, doi:10.3390/s25041113_

Round 1

Reviewer 1 Report

Comments and Suggestions for Authors

1. The abstract section emphasized the challenges of spurs and low-frequency random noise by traditional image processing. However, the introduction section gave the corresponding challenges such as internal gears, splines, cannon bores.

2. The literature review needs reorganizing because it did not emphasize the novelty and importance of the proposed method in this paper. In addition, the related references are not enough.

3. Usually, there is a space between the serial number of references and the word, such as [1, 2] in line 34. However, the style is inconsistent for other references in the introduction section.

4. Before section 2, the image acquisition process is necessary, including the utilized sensor, the collection system and the physical components.

5. Also, there are many subgraphs in Figs. 8-11. The reader may not grasp the differences and the authors’ intention immediately.

6. I do not understand the differences of the two sets of images in Fig. 12, except two different colors.

Reviewer 2 Report

Comments and Suggestions for Authors

1. This papers emphasized the necessity of high-accuracy extraction of laser stripes with complex deep-hole components. However, the application background is not clear, which reduces the necessity and urgency of this study. In addition, the lack of defining deep holes blurs the scope of application of the proposed method here.

2. Since the subsequent section did not give the sensing device, it is hard to understand the proposed ring structured light method, as well as projecting 360o structured light.

3. Two indices, DSC and HD, are used to assess the performance of the proposed method. However, there are not other methods that are used to compare these indices. The values of these indices turn to meaningless.

4. The introduction section still did not describe the specific application background.

5. In my opinion, it is not difficult to extract the laser stripe centerlines because there is weak interference in the raw image. There are many methods that have been reported to extract these centerlines from strong interference background, such as “AHP-based welding position decision and optimization for angular distortion and weld collapse control in T-joint multipass GMAW. Therefore, this paper should clearly describe the background and the main challenges. 

6. In my opinion, conclusion section is very necessary. 

Reviewer 3 Report

Comments and Suggestions for Authors

A brief summary (one short paragraph) outlining the aim of the paper, its main contributions and strengths.

This paper presents a new point cloud post-processing method based on minimum spanning tree and depth-first search to extract laser stripe centerlines from inner surfaces of complex deep holes. The method in this paper can effectively solve some traditional image processing problems, such as spurious and low-frequency random noises.  This work projects 360° structured light on the internal wall of a deep hole and uses the Steger algorithm for subpixel point cloud extraction. With MST building its connectivity structure and DFS ensuring precision during path planning and noise removal, experimental results prove that this approach has much higher precision, with a Dice Similarity Coefficient close to 1 and a small Hausdorff Distance, well outperforming previous methods. It provides a robust, flexible platform where feature extraction and 3D reconstruction of complex parts with accuracy is achievable.

Article: highlighting areas of weakness, the testability of the hypothesis, methodological inaccuracies, missing controls, etc.

Limitations of this paper include the scope of validation, as geometries tested may not fully represent all industrial deep-hole structures, hence limiting generalizability. Comparisons to other methods are not clearly made with the same dataset, which begs questions of comparison fairness. Besides, there is a lack of sufficient discussion related to the biases of the method used, missing controls, or alternative techniques for noise removal that could impact the robustness and reproducibility of the results.

Review: commenting on the completeness of the review topic covered, the relevance of the review topic, the gap in knowledge identified, the appropriateness of references, etc.

This is a highly relevant review topic that addresses the challenge of extracting accurate laser stripe centerlines associated with complex deep-hole geometries. The paper highlighted a significant gap regarding noise and spurious data processing; to a large extent, this was missing from the review, including an in-depth discussion on alternatives and their respective limitations. Generally, references appear to be appropriate, but more recent and diversified studies are needed to establish the contextual background of the presented problem. Additionally, deeper basics on the theoretical underpinning of MST and DFS in similar applications could help to develop further this review for completeness.

Specific comments referring to line numbers, tables or figures that point out inaccuracies within the text or sentences that are unclear. These comments should also focus on the scientific content and not on spelling, formatting or English language problems, as these can be addressed at a later stage by our internal staff.

Abstract. Where the results can be implemented in practice?

Introduction. The aim of this research is not clearly presented.

Figure 1. Add horizontal and vertical axis titles to (a), (c) and (d). Check all of your figures for axis titles.

Line 252. “Prim algorithm” add reference, please.

Lines 261 to 274. Make better presentation: 1-Input variables; 2-Algorithm body; 3-Output variables.

Line 287. “Depth-first search (DFS)” – add reference, please.

Line 390. “Dice Similarity Coefficient (DSC)” – add reference, please.

Line 396. It will be good to add equation numbers to all of your formulas.

Table 1. If the presented values are dimensionless, you have to describe it in the text.

It will be good to add a Conclusion part.

You have many abbreviations. It will be good to describe them in a section after the Conclusion part.

Is the manuscript clear, relevant for the field and presented in a well-structured manner?

The manuscript is clear, relevant to the field, and well-structured, effectively addressing a critical challenge in laser stripe centerline extraction for complex geometries.

Are the cited references mostly recent publications (within the last 5 years) and relevant?

The references are from 1996 to 2025.

Does it include an excessive number of self-citations?

The self-citations are reduced to minimum.

Is the manuscript scientifically sound and is the experimental design appropriate to test the hypothesis?

The manuscript is scientifically sound, with an experimental design appropriate for testing the hypothesis, though broader validation would enhance its robustness.

Are the manuscript’s results reproducible based on the details given in the methods section?

The manuscript's results appear reproducible, though additional details on dataset specifics and experimental setup would strengthen reproducibility.

Are the figures/tables/images/schemes appropriate? Do they properly show the data? Are they easy to interpret and understand? Is the data interpreted appropriately and consistently throughout the manuscript?

The figures and tables are appropriate and informative.

Are the conclusions consistent with the evidence and arguments presented?

Conclusion part is missing. The summary and a comparative analysis is made in the Discussion part.

Please evaluate the ethics statements and data availability statements to ensure they are adequate.

The authors declare no potential conflicts of interest.

The ethics and data availability statement are presented in a correct way.

Is the review clear, comprehensive and of relevance to the field? Is a gap in knowledge identified?

The review is clear, relevant, and identifies a knowledge gap, though it could be more comprehensive in covering alternative methods and their limitations.

Was a similar review published recently and, if yes, is this current review still relevant and of interest to the scientific community?

There are no papers with detailed descriptions like this one.

Novelty: Is the question original and well-defined? Do the results provide an advancement of the current knowledge?

The research question is original and well-defined, addressing a critical challenge in laser stripe extraction. The results provide a notable advancement in precision and robustness over existing methods.

Scope: Does the work fit the journal scope?

The work fits the journal's scope as it focuses on 3D sensing, optoelectronic sensors, and image sensors by leveraging structured light and laser stripe imaging for precision measurement in complex environments, contributing to advancements in sensor technology and applications in industry.

Significance: Are the results interpreted appropriately? Are they significant? Are all conclusions justified and supported by the results? Are hypotheses carefully identified as such?

The results are appropriately interpreted, significant, and well-supported by the data, with hypotheses clearly identified and justified.

Quality: Is the article written in an appropriate way? Are the data and analyses presented appropriately? Are the highest standards for presentation of the results used?

The article is well-written, with data and analyses presented clearly and to a high standard.

Scientific Soundness: Is the study correctly designed and technically sound? Are the analyses performed with the highest technical standards? Is the data robust enough to draw conclusions? Are the methods, tools, software, and reagents described with sufficient details to allow another researcher to reproduce the results? Is the raw data available and correct (where applicable)?

The manuscript is scientifically sound in its scope and focus, but its impact would be significantly enhanced by a more detailed and critical presentation of the reviewed data and methodologies.

Interest to the Readers: Are the conclusions interesting for the readership of the journal? Will the paper attract a wide readership, or be of interest only to a limited number of people? (Please see the Aims and Scope of the journal.)

The conclusions are likely to attract a wide readership, especially those interested in 3D sensing, sensor technology, and industrial applications.

Overall Merit: Is there an overall benefit to publishing this work? Does the work advance the current knowledge? Do the authors address an important long-standing question with smart experiments? Do the authors present a negative result of a valid scientific hypothesis?

Publishing this work offers clear benefits, advancing current knowledge with innovative experiments that address an important challenge in sensor technology.

English Level: Is the English language appropriate and understandable?

The English language appropriate and understandable.
